# The Relationship between Scabies and Stroke: A Population-Based Nationwide Study

**DOI:** 10.3390/ijerph16183491

**Published:** 2019-09-19

**Authors:** Meng-Huan Wu, Chien-Yu Li, Huichin Pan, Yu-Chieh Lin

**Affiliations:** 1Department of Emergency Medicine, Chang Guan Memorial Hospital, Kaohsiung 83301, Taiwan; m7169@cgmh.org.tw; 2Department of Neurosurgery, Asia University Hospital, Department of Occupational Therapy, Asia University, Taichung 41354, Taiwan; sbrain.lee@gmail.com; 3Department of Biomedical Sciences, Chung Shan Medical University, Taichung 40201, Taiwan; 4Department of Medical Research, Chung Shan Medical University Hospital, Taichung 40201, Taiwan; 5Department of Pathology and Laboratory Medicine, Taoyuan Armed Forces General Hospital, Taoyuan 32551, Taiwan; 6Department of Pathology and Graduate Institute of Pathology and Parasitology, Tri-Service General Hospital, National Defense Medical Center, Taipei 11490, Taiwan

**Keywords:** scabies, stroke, National Health Insurance Research Database

## Abstract

Background: Scabies is a commonly occurring infectious skin infestation that substantially impacts the quality of life, while stroke, which consists of a neurological deficit resulting from a lack of blood flow to the brain, carries sizable economic costs. The pathophysiologic mechanisms underlying both diseases involve inflammatory processes that are mediated by the immune system; however, no prior research has been conducted to explore the relationship between the two conditions. Methods: This population-based nationwide study utilized data from the National Health Insurance Research Database (NHIRD) of Taiwan for a total of 6628 scabies patients, who comprised a scabies group, and a randomly selected cohort of 26,509 matching patients, who served as a control group. More specifically, the medical records for the patients in both groups were checked for seven years to identify any new cases of stroke within that seven-year follow-up period. The hazard ratio (HR) of stroke for the follow-up period was then calculated using Cox proportional hazards regressions, while comorbidities and demographic characteristics were likewise analyzed. Results: During the follow-up period, 2892 patients, or 8.7%, of the overall total of 33,137 patients included in the study were newly diagnosed with a stroke. Of those newly diagnosed stroke patients, 833 were from the scabies group, and 2059 were from the control group, accounting for 12.6% and 7.8%, respectively, of the individuals in each group. With a crude hazard ratio of 1.67, the patients in the scabies group had a significantly higher risk of subsequent stroke than those in the control group, although the adjusted hazard ratio (aHR) for the scabies patients, which was determined by adjusting for covariates, was only 1.32 (95% confidence interval (CI): 1.21–1.43). Conclusions: The results of the study indicated an elevated risk of stroke among scabies patients, an association that might be contributed to by immunopathological factors. This information could serve as a reminder to clinicians to remain alert to any indications of neurological impairment in patients previously infected with scabies.

## 1. Introduction

Scabies is a highly contagious and commonly occurring ectoparasitic skin disease resulting from infection by the mite *Sarcoptes scabiei* [1,2,3]. The most common symptom is severe itchiness caused by intense pruritic skin lesions, with patient quality of life frequently being severely impacted as a result. The symptom of itchiness typically causes scratching, and the resulting scratches often lead, in turn, to secondary bacterial infections; also, an elevated danger of chronic kidney disease after scabies infection has been reported [4,5]. Immune responses and inflammatory reactions mediate the underlying pathophysiological mechanisms of scabies infection [6].

Stroke, which is also termed cerebrovascular accident (CVA), constitutes a leading source of disability around the globe, accounting for over 4% of healthcare costs in developed nations and 2% to 4% of all healthcare costs around the world [7]. A stroke subsequently leads to major disability in one-third to half of all cases, with such patients becoming heavily dependent on others for daily necessities [8]. Neurological impairments, such as dysphagia, aphasia, altered consciousness, sensory impairment, and motor weakness, constitute the most commonly occurring manifestations of stroke, while early detection of stroke risk can be helpful in allowing potential victims and their physicians to take preventative measures, with common risk factors, including obesity, diabetes, hypertension, and hyperlipidemia [9,10]. Inflammation and chronic infection have also been reported by some studies to be potential risk factors [11,12].

In spite of a potential association, no previous studies have been conducted to elucidate the relationship between stroke and scabies. The present study, a 14-year population-based study in Taiwan, was thus conducted to investigate that relationship.

## 2. Materials and Methods

### 2.1. Database

Taiwan’s National Health Insurance Research Database (NHIRD) was the source of the data used in this study. The National Health Insurance (NHI) program, which was initiated in 1995 and covered 99.9% of Taiwan’s population of 23 million as of 2013, is the national medical insurance and healthcare system of Taiwan, and the NHIRD includes the data (such as demographic data, medical records, prescription records, and medical procedures) of those who seek medical care through the NHI program [13]. More specifically, this study utilized data from the Longitudinal Health Insurance Database 2000 (LHID2000), which is a sub-database of the NHIRD and is comprised of one million randomly selected people included in the NHIRD as of 2000. This study was approved by the Institutional Review Board of the Tri-Service General Hospital (approval No: B-105-06, 04/12/2016 Taipei, Taiwan).

### 2.2. Study Population

The study population was selected from LHID2000 data covering the period from January 2000 through December 2013, with all those patients newly diagnosed with scabies between January 2001 and December 2006 comprising the scabies group (Figure 1).

The clinical diagnoses included in the NHIRD are determined by licensed physicians and are made based on the International Classification of Diseases, 9th revision, Clinical Modification (ICD-9-CM) system, with 133.0 being the ICD-9-CM code for scabies infection [14]. The exclusion criteria for the study consisted of the following: (1) any diagnosis of scabies made prior to January 1, 2001 (1420), (2) missing medical records (*n* = 115), (3) an age of 0 to 19 years old at the time of scabies diagnosis (*n* = 2492), and (4) any prior history of scabies or stroke (*n* = 1302). Ultimately, 6628 scabies patients were enrolled in the scabies group, while a non-scabies control group totaling 26,509 subjects was filled by patients randomly matched to the scabies group on the basis of age (20–29 years, 30–39 years, 40–49 years, 50–59 years, 60–69 years, and ≥70 years), gender, index year of scabies diagnosis, insured region, and urbanization at a ratio of 1:4.

Starting with his or her index date, each subject was tracked for a seven-year period to identify those who were diagnosed with stroke (ICD-9-CM: 430–438) within those seven years, with all medical diagnoses, medical procedures, and prescriptions of those subjects during the follow-up period also being identified and recorded. Each diagnosis of stroke was considered valid if made by a psychiatrist in conjunction with at least one admission to hospital or if there were at least three consistent diagnoses made in outpatient department visits, while each scabies diagnosis was determined by a licensed physician according to a detailed physical examination and history taking for the patient in question. More specifically, the clinical diagnoses of scabies required findings typical of scabies, such as inflammatory pruritic papules, burrows, or nodules; generalized itching, sparing the face; and severe pruritus, especially at night [1].

### 2.3. Covariates

The following comorbidities were selected as covariates for this study: coronary heart disease (ICD-9-CM: 410–414), congestive heart failure (ICD-9-CM: 428), chronic kidney disease (ICD-9-CM: 585,586,588), hypertension (ICD-9-CM: 401–405), hyperlipidemia (ICD-9-CM: 272.4), diabetes mellitus (ICD-9-CM: 250), and atrial fibrillation (ICD-9-CM: 427.31). Patients were categorized by age into six groups determined by ten-year intervals: 20 to 29 years, 30 to 39 years, 40 to 49 years, 50 to 59 years, 60 to 69 years, and 70 years and older. Patients were also categorized into 4 categories based on their monthly incomes in New Taiwan Dollars (NTD): less than NTD 20,000, NTD 20,000 to NTD 39,999, NTD 40,000 to NTD 59,999, and more than NTD 60,000. Patients were likewise divided into 7 categories based on the level of urbanization of their residences [15]. Finally, the patients were categorized into four groups according to the geographic area of Taiwan in which they resided: the northern region, central region, southern region, or “other” region (eastern and outlying islands).

### 2.4. Statistical Analysis

Version 19.0 of the SPSS software package (SPSS Inc., Chicago, IL, USA) was utilized for the statistical analysis, while Microsoft ^®^ SQL Server^®^ 2008 software (Microsoft, Redmond, WA, USA) was utilized for data management. Differences between the scabies group and the non-scabies control group in terms of the descriptive data, including age, income, geography, level of urbanization, and comorbidities, were analyzed using the chi-square test, while estimation of the effects of various risk factors on the hazard ratios (HRs) (with 95% CIs) was accomplished using Cox proportional hazards regression models. These models were all adjusted for the aforementioned covariates (i.e., gender, age, income, geography, urbanization, and comorbidities), and statistical significance was determined according to a two-sided *p* < 0.05.

## 3. Results

A total of 33,137 patients were enrolled in the present study, with the scabies group consisting of 6628 individuals from the NHIRD diagnosed with a scabies infection from January 2001 through December 2006. This group was compared with 26,509 control patients without scabies at a 1:4 ratio (Figure 1). There was no significant gender difference in the rate of scabies infection with younger age groups accounting for somewhat higher percentages than older age groups (Table 1). The largest percentages of the scabies patients were residents of northern Taiwan and areas of relatively high urbanization. Diabetes mellitus, hyperlipidemia, and hypertension were the most commonly occurring comorbidities. Relatedly, the scabies patients had a higher prevalence of all the comorbidities than did the participants if they were not patients in the control group.

During the seven-year follow-up period, 2892 of the total of 33,137 patients enrolled in the study (8.7%) were newly diagnosed with a stroke, including 83 from the scabies group and 2059 from the control group, accounting for 12.6% and 7.8%, respectively, of the individuals in each group (Table 2).

The two groups were thus significantly different in terms of their incidences of stroke, with the scabies patients exhibiting a substantially higher risk of subsequent stroke with a crude HR = 1.67 (95% CI: 1.54 to 1.81). A Cox regression risk analysis was conducted and indicated an adjusted HR of 1.32 (95% CI: 1.21 to 1.43) for the scabies patients (Table 3). Compared with younger patients, elderly patients exhibited a significantly higher risk of stroke (aHR = 7.99 for patients aged >70). Patients with hypertension, coronary heart disease, and atrial fibrillation were also at significantly higher risks of stroke (aHR = 3.19, 1.63, and 1.61, respectively).

## 4. Discussion

This study is, to our knowledge, the first population-based national study to investigate the relationship between scabies and stroke. The 6628 patients in the scabies group exhibited an increased risk of subsequent stroke compared to the 26,509 patients in the control group, with an adjusted HR of 1.32 for the scabies group. This finding suggested that the early and aggressive treatment of scabies might serve to lower the risk of subsequent stroke in scabies patients.

Atherosclerosis constitutes an inflammatory process that is mediated by the immune system, with vessel atherosclerosis being interacted with and accelerated by the systemic inflammation that occurs in the pathogenesis of autoimmune diseases [16,17]. Similarly, immune-mediated host inflammation after infestation by *Sarcoptes scabiei* is also found in scabies [6,15,16,17,18], along with elevated levels of interleukin (IL)-4 and IL-6 [6,14,18], the latter of which is an upstream inflammatory cytokine that has been found to play a key role in both atherosclerosis and coronary heart disease [19,20]. Moreover, fatty lesion development in mice has been found to be enhanced by IL-6, which indicates that it is a proatherogenic cytokine [21]. Studies of animal models have further indicated that IL-4 might also play a role in atherosclerosis via the induction of inflammatory responses [22,23], while atherosclerosis itself is a leading pathogenic mechanism of stroke. One of the key pathologic processes of early atherosclerosis, meanwhile, is endothelial dysfunction, which has been found to be associated with cytokine regulation [20,24]. Besides, some studies have pointed out that Pelvic Inflammation Disease (PID) is the same as rhinosinusitis, which causes atherosclerosis to produce stroke by inflammatory reaction [11,12]. Finally, according to the results of the present study, traditional cardiovascular risk factors are more prevalent in patients with scabies.

Elderly patients, who tend to have atherosclerosis, and male patients exhibited relatively high risks of subsequent stroke in this study, especially patients older than 70 years of age. This finding was not surprising, as old age is already well-known to be a major risk factor for stroke [25], with the incidence of stroke more than doubling in both men and women for every 10 years of age above 55 years old [26,27]. As noted, the results of this study were consistent with those earlier findings insofar as the patients who were older than 70 years of age had a much higher risk of stroke (aHR = 7.99, 95%CI = 6.49–9.83) than the young patients. On the other hand, while different areas have widely differing incidences of scabies, scabies has already been reported to be more common in children and adolescents than in adults in general [2,3]. The current study of adults was consistent with that insofar as scabies infections were found to be more common in young adults than in older adults, with scabies and stroke having markedly different ages of peak incidence.

Higher incidences of diabetes mellitus, hypertension, coronary heart disease, hyperlipidemia, and atrial fibrillation, all of which were found to be risk factors for stroke, were found in the scabies group than in the control group. Moreover, increased stroke risk was still found for the scabies patients even after adjustment with Cox regression analysis.

Finally, it is also worth noting that medications used to treat scabies, such as topical permethrin, lindane, benzyl benzoate, and oral ivermectin, may have stimulatory effects on the central nervous system, but no association between stroke and the use of these drugs has yet been reported.

### Limitations

The present study has several limitations. First, while the NHIRD covers approximately 99% of Taiwan’s 23 million residents, meaning that the data analyzed for this study is representative of the entire national population, the study’s cross-sectional design nonetheless means that the causal relationship, if any, between scabies and stroke cannot be directly determined. Second, the NHIRD does not include detailed laboratory test clinical data regarding patients’ functional neurologic status, nor does it include details regarding the size and location of cerebral infarcts or hemorrhages. Third, the NHIRD also lacks any data regarding individual behaviors and characteristics, such as smoking, alcohol consumption, and body mass index, that are already known to be major risk factors for stroke.

## 5. Conclusions

In summary, this population-based nationwide study of Taiwan provided evidence, demonstrating a possible connection between scabies and stroke. Specifically, an elevated risk of stroke was noted in patients with scabies, who had an adjusted HR of 1.32. Awareness of this finding may help encourage clinicians to remain alert to any signs or symptoms of stroke in patients with previous scabies infection.

## Figures and Tables

**Figure 1 ijerph-16-03491-f001:**
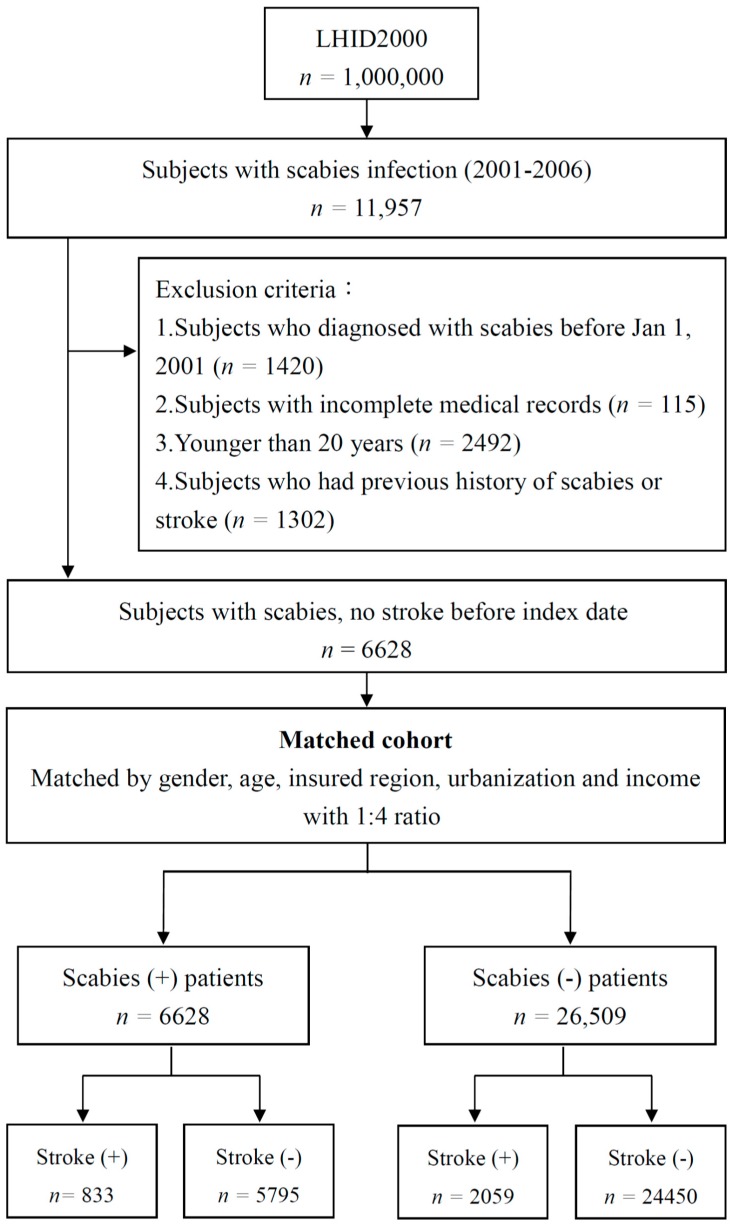
Flowchart of recruitment of subjects from the 1-million random sample of the National Health Insurance Research Database (NHIRD) from 2000 to 2006 in Taiwan. The exclusion criteria of the study was including: (1) Patients who were diagnosed with scabies before 1 January 2001 (*n* =1420); (2) Patients with missing medical records (*n* = 115); (3) patients <19 years old (*n* = 2492); (4) patients with history of scabies or stroke (*n* = 1302).

**Table 1 ijerph-16-03491-t001:** Age group, gender, and comorbidity distributions of the scabies group and control group subjects.

Variable	Number of (%) Individuals		
	With Scabies *n* = 6628	Without Scabies *n* = 26,509	*p*-Value
Gender			0.998
Female	3196 (48.2%)	12,783 (48.2%)	
Male	3432 (51.8%)	13,726 (51.8%)	
Age_Group			1
20–29	1797 (27.1%)	7188 (27.1%)	
30–39	1227 (18.5%)	4908 (18.5%)	
40–49	1236 (18.6%)	4944 (18.7%)	
50–59	869 (13.1%)	3476 (13.1%)	
60–69	552 (8.3%)	2208 (8.3%)	
≥70	947 (14.3%)	3785 (14.3%)	
Income_Group			0.001
<20,000	5039 (76%)	19,697 (74.3%)	
20,000–39,999	1027 (15.5%)	4287 (16.2%)	
40,000–59,999	433 (6.5%)	1817 (6.9%)	
≥60,000	129 (1.9%)	708 (2.7%)	
Geography			<0.001
North	3053 (46.1%)	13,893 (52.4%)	
Central	1260 (19%)	4598 (17.3%)	
South	1993 (30.1%)	7308 (27.6%)	
Other (East+ Penghu)	322 (4.9%)	710 (2.7%)	
Urbanization			<0.001
1 (highest)	1564 (23.6%)	8017 (30.2%)	
2	2051 (30.9%)	7929 (29.9%)	
3	1144 (17.3%)	4870 (18.4%)	
4	1051 (15.9%)	3440 (13%)	
5	144 (2.2%)	466 (1.8%)	
6	362 (5.5%)	947 (3.6%)	
7 (lowest)	312 (4.7%)	840 (3.2%)	
Comorbidity			
DM	1372 (20.7%)	3499 (13.2%)	<0.001
Hypertension	2274 (34.3%)	6540 (24.7%)	<0.001
CHD	1239 (18.7%)	3211 (12.1%)	<0.001
Hyperlipidemia	1645 (24.8%)	5261 (19.8%)	<0.001
CKD	464 (7%)	935 (3.5%)	<0.001
Atrial_fibrillation	176 (2.7%)	375 (1.4%)	<0.001

DM, Diabetes mellitus. CHD, coronary heart disease. CKD, chronic kidney disease.

**Table 2 ijerph-16-03491-t002:** Cox regression analysis results regarding the degree to which a past scabies infection is predictive of subsequent stroke versus no such past infection.

Variable	Number of (%) Individuals	
	With Scabies *n* = 6628	Without Scabies *n* = 26,509
With stroke	833 (12.6)	2059 (7.8)
Without stroke	5795 (87.4)	24,450 (92.2)
Crude HR	1.67 (1.54–1.81) ^‡^	-

^‡^*p* < 0.001 for comparison between patients in the two groups. HR, hazard ratio.

**Table 3 ijerph-16-03491-t003:** Cox regression analysis results regarding the degree to which independent factors are predictive of stroke.

Variable	Crude	Adjusted
	HR (95% CI)	HR * (95% CI)
Scabies	1.67 (1.54–1.81) ^‡^	1.32 (1.21–1.43) ^‡^
Gender		
Female	1	-
Male	1.06 (0.99–1.14)	-
Age_Group		
20–29	1	1
30–39	2.01 (1.58–2.57) ^‡^	1.69 (1.32–2.16) ^‡^
40–49	4.4 (3.56–5.46) ^‡^	2.63 (2.11–3.28) ^‡^
50–59	9.51 (7.74–11.68) ^‡^	3.74 (3.01–4.65) ^‡^
60–69	18.49 (15.1–22.64) ^‡^	5.53 (4.46–6.87) ^‡^
≥70	24.32 (20.05–29.52) ^‡^	7.99 (6.49–9.83) ^‡^
Income_Group		
<20,000	1	1
20,000–39,999	0.54 (0.47–0.6) ^‡^	1.02 (0.9–1.17)
40,000–59,999	0.44 (0.36–0.54) ^‡^	0.77 (0.62–0.94) ^†^
≥60,000	0.58 (0.43–0.76) ^‡^	0.72 (0.54–0.96) ^†^
Geography		
North	1	1
Central	1.13 (1.02–1.25) ^‡^	0.94 (0.84–1.05)
South	1.23 (1.13–1.34) ^‡^	0.96 (0.87–1.05)
Other (East+ Penghu)	1.73 (1.45–2.06) ^‡^	1.03 (0.86–1.25)
Urbanization		
1 (highest)	1	1
2	1.07 (0.97–1.19)	1.04 (0.94–1.16)
3	1.2 (1.07–1.35) ^†^	1.14 (1.01–1.28) ^†^
4	1.61 (1.44–1.81) ^‡^	1.15 (1.01–1.29) ^†^
5	2.31 (1.86–2.86) ^‡^	1.18 (0.94–1.47)
6	1.84 (1.55–2.17) ^‡^	1.07 (0.9–1.29)
7 (lowest)	1.85 (1.55–2.21) ^‡^	1.13 (0.94–1.37)
Comorbidity		
DM	4.44 (4.12–4.78) ^‡^	1.36 (1.25–1.47) ^‡^
Hypertension	9.4 (8.64–10.23) ^‡^	3.19 (2.88–3.54) ^‡^
CHD	6.25 (5.81–6.73) ^‡^	1.63 (1.5–1.77) ^‡^
Hyperlipidemia	3.09 (2.87–3.32) ^‡^	1.16 (1.07–1.26) ^‡^
CKD	4.89 (4.42–5.41) ^‡^	1.25 (1.13–1.39) ^‡^
Atrial_fibrillation	7.36 (6.47–8.38) ^‡^	1.61 (1.4–1.84) ^‡^

* Each variable was adjusted for every other variable listed whose crude HR was significant (*p* < 0.05). ^†^
*p* < 0.05 for comparison between patients with two groups. ^‡^
*p* < 0.001 for comparison between patients with two groups.

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
