# Peer review of "The Relationship between Scabies and Stroke: A Population-Based Nationwide Study"

_ijerph, 2019, doi:10.3390/ijerph16183491_

Round 1
Reviewer 1 Report
Two other studies have been conducted in Taiwan looking at stroke and PID and rhinosinusitis, had you considered either of these to be included in your analysis?
Some minor edits attached for your consideration.

Author Response
Responses to Reviewer 1
Two other studies have been conducted in Taiwan looking at stroke and PID and rhinosinusitis, had you considered either of these to be included in your analysis?
Response: Thanks for your comment. About the use of the National Health Insurance Research Database (NHIRD) of Taiwan to analyze the relationship between stroke and Pelvic Inflammation Disease (PID) or stroke and rhinosinusitis. These studies have been published by a team led by Dr. Hui-Wen Lin, 1. Chen PC, Tseng TC, Hsieh JY, Lin HW. Association between stroke and patients with pelvic inflammatory disease: a nationwide population-based study in Taiwan .2. Wu CW, Chao PZ, Hao WR, Liou TH, Lin HW. Risk of stroke among patients with rhinosinusitis: a population-based study in Taiwan. Am J Rhinol Allergy. 2012 Jul-Aug;26(4):278 -82.
We also put the results of these two articles into our discussion. Please see the discussion page 7 line 5-7
Some minor edits attached for your consideration.
Response: Thanks for your comment. We have modified it according to your suggestions.

Reviewer 2 Report
Thank you for the opportunity to review this manuscript. It explores an original hypothesis, that scabies infestation is related to the incidence of stroke. The authors conduct a cohort study with matched unexposed subjects and find a moderately strong association that is attenuated with adjustment for potential confounders, but still statistically significant.
My main problem with the manuscript is that the authors do not detail how they conducted the analysis has not taken account of the matching in the study design. The authors state that they have used a Cox model, however, in this situation, it is usual to stratify (estimate the same baseline hazard) on the matched group. Redoing the analysis and accounting for the clustering would produce more convincing results. Further unmatched covariates may be adjusted for.
It is also not clear when the covariate information was recorded. This should be before enrolment in the cohort rather than during follow-up.
It is clear that the scabies group has higher proportions of subjects with risk factors for cardiovascular disease (diabetes, prior CVD) than the matched group. Nevertheless, it is an interesting hypothesis that could be tested in other populations.
It would also be interesting to know whether or not the risk of overall mortality is different in the 'scabies treated' compared to the 'no scabies' population.
Author Response
Responses to Reviewer 2
1.My main problem with the manuscript is that the authors do not detail how they conducted the analysis has not taken account of the matching in the study design. The authors state that they have used a Cox model, however, in this situation, it is usual to stratify (estimate the same baseline hazard) on the matched group. Redoing the analysis and accounting for the clustering would produce more convincing results. Further unmatched covariates may be adjusted .
Response: Thanks for your comment. We have adjusted in table3 for pairing conditions such as age, insuned region, urbanization and income. Please see table3
2.It is also not clear when the covariate information was recorded. This should be before enrolment in the cohort rather than during follow-up.
Response: Thanks for your comment. In this study we used the LHID2000 database. LHID2000 contains all the original claim data of 200,000 individuals randomly sampled from the 2000 Registry for Beneficiaries (ID) of the NHIRD, which maintains the registration data of everyone who was a beneficiary of the National Health Insurance Program. There are approximately 23.72 million individuals in this registry. All the registration and claim data of these 200,000 individuals collected by the National Health Insurance program during the period of 1996-2013 constitute the LHID2000.
3.It is clear that the scabies group has higher proportions of subjects with risk factors for cardiovascular disease (diabetes, prior CVD) than the matched group. Nevertheless, it is an interesting hypothesis that could be tested in other populations.
Response: Thanks for your suggestion. We will put your suggestions in our future research.
4.It would also be interesting to know whether or not the risk of overall mortality is different in the 'scabies treated' compared to the 'no scabies' population.
Response: Thanks for your comment. Death-related analysis is a limitation in our research. National Health Insurance Research Database of Taiwan has strict restrictions on the eligibility of death data, and we are not given the right to obtain relevant information.
